# Calibration Methods for Large-Scale and High-Precision Globalization of Local Point Cloud Data Based on iGPS

**DOI:** 10.3390/s24186114

**Published:** 2024-09-21

**Authors:** Rui Han, Thomas Dunker, Erik Trostmann, Zhigang Xu

**Affiliations:** 1Fraunhofer Institute for Factory Operation and Automation IFF, Sandtorstrasse 22, 39106 Magdeburg, Germany; 2Shenyang Institute of Automation, Chinese Academy of Sciences, Nanta Street 114, Shenyang 110016, China; zgxu@sia.cn; 3University of Chinese Academy of Sciences, Zhongguancun East Road 80, Beijing 100049, China

**Keywords:** generalized hand–eye calibration, indoor global positioning system (iGPS), local data globalization, point cloud registration, optical metrology, multiple sensor data fusion

## Abstract

The point cloud is one of the measurement results of local measurement and is widely used because of its high measurement accuracy, high data density, and low environmental impact. However, since point cloud data from a single measurement are generally small in spatial extent, it is necessary to accurately globalize the local point cloud to measure large components. In this paper, the method of using an iGPS (indoor Global Positioning System) as an external measurement device to realize high-accuracy globalization of local point cloud data is proposed. Two calibration models are also discussed for different application scenarios. Verification experiments prove that the average calibration errors of these two calibration models are 0.12 mm and 0.17 mm, respectively. The proposed method can maintain calibration precision in a large spatial range (about 10 m × 10 m × 5 m), which is of high value for engineering applications.

## 1. Introduction

One of the most important developments of Industry 4.0 is digitization. Traditional manufacturing and assembly environments are actively undergoing digital transformation, hoping to further improve their competitiveness in terms of capacity, efficiency, product quality, and cost reduction through digitization. Many related research areas have emerged, including digital twin technology [1], shop-floor scheduling planning [2], re-configurable manufacturing systems [3], and so on. High-accuracy measurement is one of the important foundations for realizing industry digitization because of its advantages, such as accurate positioning for machining, improving machining accuracy, physical time-based modeling, realizing reverse engineering, the optimization of work processes based on big data algorithms, improving production quality and efficiency, and so on.

The 3D scanner is one of the most widely used local measurement devices in the industrial field, which can directly generate accurate three-dimensional point cloud data. It is widely used in the fields of reverse modeling, topographic inspection, and quality defect inspection. As the most important tool carrier, the industrial robot has the advantages of high flexibility, a large operating range, and high positioning accuracy. Therefore, 3D scanners and industrial robots are combined in many applications. The 3D scanner is installed at the end of the robot, which not only expands the measurement range but also accurately locates the local measurement position, allowing the 3D scanner to be used more widely. In some applications, engineers and researchers also combine the robot with a mobile platform to further increase the measurement range of the 3D scanner. However, since the FOV (field of view) of the 3D scanner is generally small, when using a 3D scanner to measure large components, it is necessary to take many different local measurements and transform each measurement from the local coordinate system (3D scanner coordinate system) to the global coordinate system, and then stitch the results of multiple measurements to obtain the final measurement results. Therefore, it is particularly important to calculate the transformation matrix between the local and global coordinate systems with high precision in real time. Even if the 3D scanner provides high measurement accuracy, if the accuracy of the transformation matrix is not high enough, the final result still cannot be maintained at a high accuracy level. Because the point cloud data acquired by the 3D camera need to be transformed into the global coordinate system in real time, an external measurement device is needed to measure and calculate the transformation matrix between the local and global coordinate systems.

The laser tracker is one of the most widely used 3D coordinate measuring devices in industrial applications with advantages such as high accuracy, a large measuring range, and excellent dynamic tracking performance. A large number of researchers choose the laser tracker as an external measurement device to accomplish the transformation of point cloud data from the local coordinate system into the global coordinate system. Z. Chen et al. [4] used a laser tracker with MAXscan to acquire point cloud data. Since the laser tracker can only track a single target, the authors needed to perform calibration before each measurement. A. Paoli et al. [5] used a 3D scanner, a robot, a laser tracker, and an orbital moving platform to measure large yacht components. They first used a checkerboard grid to calibrate the 3D scanner coordinate system with the robot TCP (tool center point) coordinate system (hand–eye calibration) and then transformed the 3D data into the platform coordinate system using the joint parameters provided by the robot and the positional data provided by the orbital moving platform. A laser tracker was used to calibrate the platform coordinate system with the global coordinate system, finally realizing the transformation of 3D data from the local coordinate system into the global coordinate system. The obvious problem with this method is that it introduces errors when using the robot joint data and moving platform position data, leading to a decrease in calibration accuracy. Similarly, Z. Zhou et al. [6] used a laser tracker, a 3D scanner, a robot, and a moving platform to measure large components. The difference is that they mounted the SMR (spherically mounted retro-reflector) on the 3D scanner and designed a calibration target consisting of the SMR and a standard sphere to achieve direct calibration of the 3D scanner coordinate system with the laser tracker’s coordinate system. This method avoids errors introduced when using robot joint data; however, the laser tracker can only track one target at a time and has a large tracking dead zone when used with the SMR. The authors had to design complex calibration targets and algorithms to realize the direct transformation from the 3D scanner coordinate system into the laser tracker coordinate system. Through these two studies, it is evident that although the laser tracker has the advantages of high accuracy and a large measuring range, it is not suitable as an external measurement device to transform the local point cloud data into the global coordinate system in real time because of its limited number of tracking targets and large tracking dead zone.

Some researchers use other external measurement devices to accomplish this goal. J. Wang et al. [7] used a mobile robot, a 3D scanner, and a stereo vision system to transform the local point cloud data into the global coordinate system. They first used coded marks to calibrate the 3D scanner coordinate system with the stereo vision coordinate system and then used the coded marks on the ground to calibrate the stereo vision coordinate system with the global coordinate system to complete the transformation of local point cloud data into the global coordinate system without introducing errors from the robot’s internal parameters. Later, J. Wang et al. [8] proposed another similar method by using a laser positioning sensor with higher positioning accuracy to compute the transformation matrix between the mobile platform coordinate system and the global coordinate system, but this time, they used the robot joint parameters for hand–eye calibration, which introduced some errors. M. Lai et al. [9] used an optical tracking system as an external measurement device and a checkerboard grid to calibrate the 3D scanner coordinate system with the optical tracking coordinate system. Conversely, S. Yin et al. [10] chose not to use external measurement equipment but only a 3D scanner, a robot, high-precision linear guides, and a known-position calibration sphere to achieve the transformation of local point cloud data into the global coordinate system. Problems with this method include the introduction of errors from the robot joints and linear guides, and the measurement range is not large enough.

This paper focuses on the large-scale and high-accuracy globalization of local point cloud data and first proposes the use of an iGPS (indoor Global Positioning System) as an external measurement device for local data globalization. In addition to the advantages of high measurement accuracy and a large measurement range, more importantly, an iGPS can measure multiple targets at the same time. Leveraging these advantages, the establishment of multiple sensor coordinate systems can be executed simultaneously to realize the transformation of a large number of point cloud data from different devices into the global coordinate system at the same time. This paper discusses in detail how to integrate the iGPS sensor with a 3D scanner into a measurement unit, how to construct the iGPS sensor coordinate system and calibrate the 3D scanner coordinate system with the sensor coordinate system, and finally, verify the accuracy of the transformation through experiments.

This paper consists of five main sections. Section 1 describes the importance of researching the large-scale and high-accuracy globalization of local point cloud data and presents the contributions made by other researchers while also pointing out their shortcomings. Section 2 briefly introduces the components, features, and advantages of the iGPS, as well as two methods for using iGPS sensors to establish a sensor coordinate system. For the varying conditions of different applications, two different calibration methods are given in Section 3. Section 4 describes the specific experimental procedures and steps for calibration tests based on the two calibration principles. Section 5 demonstrates the validity of these methods through verification experiments and compares the features with research by other authors to prove the superiority of this method. Finally, Section 6 presents the conclusions of this paper and the future work.

## 2. About the iGPS

### 2.1. Brief Introduction to the iGPS

The iGPS is a widely used industrial measurement system. This measurement system was developed in the late 1990s under the name Constellation 3DI by a corporation named Metris. In November 2009, Metris was acquired by Nikon Metrology NV. Now, the iGPS team has formed an independent company in Canada called 7D Metrology [11,12]. The iGPS is a 6 DOF (Degree of Freedom) laser-based measurement system based on a triangulation positioning algorithm that measures position and orientation [13]. The main components of the iGPS are two or more transmitters, a control center, and several wired/wireless receivers [14], as shown in Figure 1.

The operating transmitter emits strobe impulses and two rotating laser planes. The receivers determine position and orientation using these laser signals. The time difference between the three laser signals is used to calculate the elevation angle and azimuth angle between the transmitter and receiver. At least two line-of-sight transmitters are required to determine the position of a receiver. Using more than two transmitters can minimize the risk of covering lines of sight for some positions in the measuring volume and improve the accuracy of the measurement, as shown in Figure 2 and Figure 3 [15,16,17,18,19,20].

Based on this structural composition and working principle, the iGPS has many advantages, including a large measurement range, high measurement accuracy, high scalability, cost-effectiveness, and no reception dead space in the receiver. One of the unique advantages of the iGPS is that it can measure and track multiple targets at the same time and can measure position and orientation at the same time with sub-millimeter accuracy [21,22,23]. This advantage is unmatched by any other industrial measurement device and is the reason why the iGPS was chosen as the measurement device for this research.

### 2.2. Establishment of the iGPS Sensor Coordinate System

The iGPS receiver chosen for this study is the i5IS, as shown in Figure 1. The two black metal parts (marked by the red dotted box) are used to receive the signals emitted by the transmitter, which are the direct measurement points of the i5IS. Since the iGPS can measure multiple targets at the same time, these two points can be measured simultaneously.

After installing two i5IS sensors together in suitable positions relative to each other, the coordinates of four points (two points from each i5IS sensor) can be measured and recorded at the same time. This subsection introduces two methods for establishing the sensor coordinate system using three points or four points, and both of them need to use two i5IS sensors, along with the right-hand coordinate system principle, as shown in Figure 4.

The first method uses any three of the four points. The steps of this method are as follows:Select three points on the two i5IS sensors and name them points A, B, and C. It is recommended to choose the three points that can form the largest triangle to maximize the precision of establishing the coordinate system. Determine the plane α formed by points A, B, and C, as shown in Figure 5a.Any point (A) among A, B, and C is chosen as the origin of the sensor coordinate system. The normal vector of the plane α is calculated and defined as the *z*-axis of the sensor coordinate system, as shown in Figure 5b.Choose any point (B) other than the origin, and connect points A and B to form the vector $\vec{AB}$, which is the *x*-axis of the sensor coordinate system, as shown in Figure 5c.Compute the cross-product of the existing two vectors representing the *x*-axis and *z*-axis; the resulting vector is defined as the *y*-axis of the sensor coordinate system, as shown in Figure 5d.

The second method uses all four points. The steps of this method are as follows:Measure all four points at the same time and name them points A, B, C, and D. Create the vector $\vec{v_{1}}$ with points C and D, and create the vector $\vec{v_{2}}$ with points A and B. Find the shortest line segment between the two vectors and define the midpoint of the line segment as the origin of the sensor coordinate system, as shown in Figure 6a.Determine the *z*-axis of the sensor coordinate system by the direction of the shortest line segment, as shown in Figure 6b.Determine the *x*-axis of the sensor coordinate system by the origin and the direction of the vector $\vec{v_{1}}$ (or vector $\vec{v_{2}}$), as shown in Figure 6c.Compute the cross-product of the existing vectors representing the *x*-axis and *z*-axis, and use the right-hand coordinate system principle to determine the direction; the resulting vector is defined as the *y*-axis of the sensor coordinate system, as shown in Figure 6d.

Readers can choose the appropriate method to establish the sensor coordinate system according to their own needs, as both methods can establish the sensor coordinate system with high precision. Based on these two methods for establishing the coordinate system, this research integrates two i5IS sensors with the 3D scanner chosen for this study, the LMI GOCATOR3210, to form the integrated measurement unit, as shown in Figure 7.

## 3. Calibration Principle

As previously mentioned, the laser tracker is a widely used industrial device. While calibrating the iGPS coordinate system, the laser tracker can be used to measure the fixed stable points in advance, which not only improves the accuracy of the calibration but also naturally aligns the iGPS coordinate system with the laser tracker coordinate system. However, due to the high cost of the laser tracker, not all enterprises, research organizations, and laboratories can conveniently use it. Therefore, two calibration schemes and corresponding solutions are given in this research, depending on whether a laser tracker is used.

The notations are given first to help readers understand easily. For the method with a laser tracker, two different coordinate systems of the iGPS and laser tracker are aligned first, so they can both be considered as the global coordinate system, denoted as **{G}**. The 3D scanner coordinate system is denoted as **{C}**, the i5IS sensor coordinate system is denoted as **{S}**, and the workpiece coordinate system is denoted as **{P}**. The notation TABi is used to denote the transformation matrix from coordinate system **{A}** to **{B}** at the *i*-th measurement.

### 3.1. Method with a Laser Tracker

By using a laser tracker for calibration, the calibration principle can be constructed, as shown in Figure 8, and Equation (Equation 1) can be used to express the coordinate transformation relationship.
(1)MRiG=TSGiTCSMRiC
where $M\;R_{i}^{C}$ and $M\;R_{i}^{G}$ denote the measurement results of the 3D scanner and the laser tracker, respectively, in the *i*-th measurement. TSGi can be calculated and recorded in real time from the measurement results of the i5IS sensor, and $T_{C}^{S}$ is the transformation matrix that needs to be calibrated and calculated. The problem of solving $T_{C}^{S}$ can be described as an optimization problem, as shown in Equation (Equation 2).
(2)minTCS∑i=1nMRiG−TSGiTCSMRiC2

A large number of scholars and researchers have already investigated the solution to such optimization problems, among which the most classical methods include the least-squares method proposed by K. S. Arun et al. [24], the orthogonal-matrices-based method proposed by Berthold K. P. Horn et al. [25], the quaternions-based method proposed by Berthold K. P. Horn et al. [26], and the method of least-squares estimation of the transformation parameters proposed by S. Umeyama [27]. It is worth noting that the laser tracker is only used during the calibration stage and is no longer needed after calibration is completed.

### 3.2. Method without a Laser Tracker

In order to ensure this research can help different applications, this subsection shows how to calibrate without a laser tracker. The calibration principle can be constructed as shown in Figure 9, and the coordinate system transformation relationship is obtained as shown in Equation (Equation 3).
(3)TSGjTGP=TSCTCPj

In this principle, the position of the workpiece in space is constant and stable, while the position of the 3D scanner is adjusted several times for multiple measurements. TSGj and TCPj can be directly calculated through the measurements from the i5IS sensor and the 3D scanner, and $T_{G}^{P}$ and $T_{S}^{C}$ are the constant and to-be-solved matrices in the calibration model. Then, another common mathematical model in the field of calibration is acquired: $A_{j}\;X\;=\;Y\;B_{j}$. Similarly, there has been a large amount of research on solving this problem. The most classical solutions include the method based on the double quaternions and the Kronecker product proposed by L. Aiguo et al. [28], the method based on the quaternion algebra and positive quadratic error function proposed by F. Dornaika et al. [29], the method based on the Kronecker product proposed by M. Shah. [30], and the linear solution method proposed by H. Zhuang et al. [31].

## 4. Calibration Experiments

### 4.1. Experiment with a Laser Tracker

A calibration experiment with a laser tracker is conducted first to realize the accurate globalization of local point cloud data. According to the first calibration principle, the key step in realizing accurate globalization is the high-accuracy calibration of the 3D scanner coordinate system with the i5IS sensor coordinate system. The hardware used in this calibration experiment includes a Leica ATS600 laser tracker with a 0.5-inch SMR (spherically mounted retro-reflector); an iGPS from 7D Metrology with three i5IS sensors, four transmitters, and four monuments; a Gocator3210 3D scanner from LMI; a YASKAWA HC20DT robot; and a calibration board with four standard spheres with known radii. The configuration of this calibration experiment is shown in Figure 10.

The detailed calibration process is described in the following steps:Calibrate the iGPS with the laser tracker and align these two coordinate systems. In this step, SMRs are placed on the top magnetic base of all four monuments, and the laser tracker is used to measure the positions of these monuments. Based on these measurements, two different coordinate systems can be aligned precisely and automatically.The integrated measurement unit is mounted at the end of the robot, and the calibration board is placed in a suitable position. Figure 11 shows the relative position between the calibration board and the robot with the integrated measurement unit.Next, the SMR of the laser tracker is used as a probe to perform the contact continuous measurement of one standard sphere on the board. The measuring process is shown in Figure 12, and the result is shown in Figure 13. The coordinate of the sphere’s center in the global coordinate system is fitted and calculated by the measurements and Equation (Equation 4), where $c_{\mathrm{ss}}^{G}$ is the coordinate of the sphere’s center in the global coordinate system, cPGi is the coordinate of the valid points from the laser tracker, *n* is the number of valid points, $R_{ss}$ is the radius of the standard sphere, and $R_{S\;M\;R}$ is the radius of the SMR.
(4)mincssG∑i=1ncssG−cPGi2−Rss+RSMR2Control the robot carrying the integrated measurement unit to perform the measurement of the same sphere several times from several different positions, and record the measurement from the 3D scanner and the i5IS sensor simultaneously. The measuring process and the measurement result are shown in Figure 14 and Figure 15, respectively.Fit and calculate the coordinates of the sphere’s center in the 3D scanner coordinate system through the measurements and Equation (Equation 5):
(5)mincssC∑i=1mcssC−cPCj2−Rss2
where $c_{\mathrm{ss}}^{C}$ represents the coordinates of the sphere’s center in the 3D scanner coordinate system, cPCj represents the coordinates of one of the valid points in the 3D scanner coordinate system, *m* is the number of valid points from the 3D scanner, and $R_{ss}$ is the radius of the standard sphere. The sphere’s center coordinates in the 3D scanner coordinate system after calculating and fitting are integrated into a matrix, as shown in Equation (Equation 6).
(6)CssC=cssC1cssC2…cssCk3×k
where $C_{\mathrm{ss}}^{C}$ denotes the coordinate matrix of the sphere’s center in the 3D scanner coordinate system. The dimension of the matrix is 3×k, with *k* representing the number of multiple measurements from different positions.Each time the position of the integrated measurement unit is adjusted, the data from the i5IS sensor are also recorded to transform the rotation matrix $r_{G}^{S}$ and the translation vector $t_{G}^{S}$ from the global coordinate system into the sensor coordinate system. Integrate *k*-times measurements of the rotation matrices and translation vectors, as shown in Equation (Equation 7):
(7)RGS=rGS1rGS2…rGSk3×3×kTGS=tGS1tGS2…tGSk3×kBased on the calculation above, the coordinates of the sphere’s center in the global coordinate system are transformed into the sensor coordinate system, as shown in Equation (Equation 8):
(8)$$C_{\mathrm{ss}}^{S}={\left[R_{G}^{S}\right]}^{T}\;c_{\mathrm{ss}}^{G}+T_{G}^{S}$$
where $C_{\mathrm{ss}}^{S}$ is the coordinate matrix of the sphere’s center in the sensor coordinate system with dimensions of 3×k.Based on the calculation above, $R_{C}^{S}$ and $T_{C}^{S}$ are calculated according to Equation (Equation 9):
(9)$$\min_{R_{C}^{S},T_{C}^{S}}∥C_{\mathrm{ss}}^{S}-\left(R_{C}^{S}\;C_{\mathrm{ss}}^{C}+T_{C}^{S}\right)∥$$

The flowchart of the whole calibration experiment process is shown in Figure 16.

### 4.2. Experiment without a Laser Tracker

This subsection describes the process and methodology of calibrating the 3D scanner coordinate system with the sensor coordinate system without a laser tracker. The devices used in this calibration experiment are essentially the same as those in the previous experiment, with the only difference being that a laser tracker is no longer used, as shown in Figure 17. The detailed calibration process is described in the following steps:Calibrate the iGPS and align it with the global coordinate system.Mount the integrated measurement unit at the end of the robot.In this calibration model, the target coordinate system needs to be established. Therefore, three standard spheres need to be measured by the 3D scanner simultaneously, and the coordinates of these three sphere centers need to be calculated. Move the robot carrying the integrated measurement unit to different positions to measure, as shown in Figure 18, and record the measurements, as shown in Figure 19.Based on the calculation above, the target coordinate system can be constructed using the same method mentioned previously, as shown in Figure 20.Then, the transformation matrix from the target coordinate system to the 3D scanner coordinate system for each measurement is calculated and integrated into a matrix, as shown in Equation (Equation 10):
(10)TOC=TOC1TOC2…TOCk4×4×k
where $T_{O}^{C}$ is the integrated matrix with dimensions of $4\times \left(4\times k\right)$, where k is the number of multiple measurements.During the multiple measurements, the data from the i5IS sensor are also recorded to calculate the transformation matrix from the global coordinate system to the sensor coordinate system for each measurement. The integration matrix is shown in Equation (Equation 11):
(11)TGS=TGS1TGS2…TGSk4×4×k
where $T_{G}^{S}$ is the integrated matrix with dimensions of $4\times \left(4\times k\right)$.Based on the above data, the transformation matrices $T_{C}^{S}$ and $T_{O}^{G}$ are calculated using Equation (Equation 12):
(12)minTCS,TOG∑q=1kTOCqTCS−TOGTGSq

The flowchart of the calibration process without using a laser tracker is shown in Figure 21.

## 5. Verification Experiments

In this section, we validate the accuracy of the calibration results. We provide a description of the validation experiments, an analysis of the experimental results, and a comparison of the accuracy with current state-of-the-art methods.

### 5.1. Verification Experiment I

In order to verify the accuracy of the calibration method proposed in this paper, the measurements from the laser tracker are chosen as a reference because of their higher accuracy. We first align them into one coordinate system and then place an SMR on the magnetic base of the i5IS sensor, as shown in Figure 22. We then position the i5IS sensor with the SMR at different locations in the experimental space and measure the target with the iGPS and laser tracker simultaneously. To check the alignment error between these two coordinate systems, we place the target at several different positions and measure using the two devices. The alignment errors at different positions are shown in Table 1.

After ensuring the alignment accuracy, we place the calibration board at an arbitrary position in the measurement field and measure the standard sphere directly using the laser tracker with the SMR (this process is similar to the configuration shown in Figure 12) and the integrated measurement unit (this process is similar to the configuration shown in Figure 11). When measuring the standard sphere using the laser tracker with the SMR, we can fit the coordinates of the sphere’s center in the global coordinate system directly. When measuring the standard sphere using the integrated measurement unit, we first fit the coordinates of the sphere’s center in the 3D scanner coordinate system through the point cloud obtained from the 3D scanner and then transfer these coordinates to the global coordinate system using the transformation matrix calibrated previously. The calibration error is quantified by comparing the Euclidean distance of the coordinates of the sphere’s center in the global coordinate system, measured using the two measurement methods, as shown in Equation (Equation 13):(13)$$\varepsilon_{Trans}={∥c_{SSC}^{LT}-c_{SSC}^{C-iGPS}∥}_{2}$$
where $c_{SSC}^{LT}$ represents the coordinates of the sphere’s center measured using the laser tracker, $c_{SSC}^{C-iGPS}$ represents the coordinates of the sphere’s center measured using the integrated measurement unit, and $\varepsilon_{Trans}$ is the calibration error.

To verify that the method guarantees calibration accuracy in the full measurement space, we place the calibration board at several different positions in the measurement space, as shown in Figure 23, repeat the verification above, and calculate the error at these different positions.

Two calibration methods are performed. The calibration errors are shown in Figure 24, while the average and maximum errors of the measurement results are shown in Table 2.

In Table 2, M-A indicates the measurements from the integrated measurement unit using the calibration method with the laser tracker, while M-B indicates the measurements from the integrated measurement unit using the calibration method without the laser tracker. From the results in Figure 23 and Table 2, it is evident that the method using the laser tracker has higher accuracy, but the 3D errors for both methods are less than 0.1 mm, which meets the general accuracy requirements for industrial applications. Also, it is evident that calibration accuracy is not affected by changes in the measurement position.

### 5.2. Verification Experiment II

The previous experiment assumes that the measurements from the laser tracker are the theoretical true values, which may not be accurate enough. In addition, the alignment error between the laser tracker and the iGPS is included. Therefore, another verification experiment is designed, and the configuration is shown in Figure 25.

The calibration board is placed at positions 5 to 6 m and 12 to 14 m away from the laser tracker. It is important to ensure that the laser tracker and these two positions are successively aligned in a straight line, which minimizes the rotation of the tracker’s head and reduces the error introduced by this mechanical movement. In order to avoid errors caused by the alignment of the two coordinate systems, we quantified the calibration error through the absolute distance between the two points instead of the absolute coordinates of one point in the experimental space. The two positions of the robot in this experiment are shown in Figure 26.

There are four standard spheres on the calibration board. When the calibration board is at position I, the positions of the four standard spheres are designated as A, B, C, and D. When the calibration board is at position II, the positions of the four standard spheres are designated as A’, B’, C’, and D’. The line connecting the centers of any two standard spheres at position I and position II can be used as a single measurement, as shown in the blue box in Figure 25. The measurements include A-A’, A-B’, …, D-C’, and D-D’, for a total of 16 measurements. These 16 spatial distances are measured using the laser tracker and the integrated measurement unit, respectively, and the measurements are shown in Table 3.

In Table 3, LT indicates the measurements from the laser tracker. The measurements from the laser tracker are again considered the theoretical true values and the calibration error is quantified by calculating the differences between the verification method and the theoretical true value. The calibration errors of the two methods in this experiment are shown in Figure 27, and the statistical characteristics of the calibration errors are shown in Table 4.

Based on the results of this verification experiment, it is evident that the accuracy of the measurement results using the two calibration methods discussed in this paper can still be maintained at a high level in a large-scale measurement volume and meet the requirements of industrial applications, after further improving the reliability of the measurements from the laser tracker as the theoretical truth value.

### 5.3. Verification Experiment III

To further validate that the methods discussed in this paper are able to accurately globalize local point cloud data with high flexibility, we designed Experiment III based on Experiment II. In Experiment III, both the robot and the laser tracker are positioned in the same way as in Experiment II, and the object being measured is changed to an independent standard sphere with a tripod as the base. During the measurement, the robot carries the integrated measurement unit to measure the standard sphere in completely different relative postures, as shown in Figure 28.

The overall configuration of Experiment III is shown in Figure 29, where the integrated measurement unit measures the standard sphere in four completely different measurement postures, denoted as E, F, G, and H at position I and E’, F’, G’, and H’ at position II.

The coordinates of the spheres’ centers are calculated according to the different measurements separately, and the absolute distances between the spheres’ centers at the two positions are used to quantify the calibration error. The measurements from the laser tracker are used as a reference. The calibration errors of different combinations of measurement postures with the two calibration methods are shown in Table 5.

Similarly, the error line plots and statistical error characteristics of M-A and M-B in Experiment III are shown in Figure 30 and Table 6.

By comparing the results of Experiment II and Experiment III, it is evident that the results from these two experiments are extremely similar, regardless of the maximum error, average error, or standard deviation of the error. This further proves that the methods discussed in this paper possess high flexibility for local data globalization in a large-scale measurement volume.

### 5.4. Comparison of Methods

In order to ensure that the methods discussed in this paper demonstrate sufficient superiority, they are compared with current state-of-the-art methods. The features being compared include the average error, maximum error, and the scale of the measurement volume. The comparison data are shown in Table 7.

From this comparison, it is evident that the two methods discussed in this paper can both globalize local point cloud data with high flexibility in a large measurement volume. Thanks to the advantages of the iGPS, the sensor coordinate system can be accurately established, allowing local point cloud data to be directly transformed into the global coordinate system. This eliminates the need for complex intermediate coordinate system transformations, reduces the introduction of errors, and improves accuracy.

It is worth noting that both methods for establishing the sensor coordinate system mentioned in Section 2.2 were used in the calibration and validation experiments. A large number of experiments have proved that both methods can achieve the goal of establishing a high-precision sensor coordinate system, and researchers can choose either of these methods based on their specific hardware conditions and application environments.

## 6. Conclusions and Future Work

In this paper, a new local point cloud data globalization method based on the iGPS is proposed. Considering the different conditions of hardware preparation, two calibration methods (with and without a laser tracker) and their principles are described separately. Three different verification experiments are conducted sequentially to demonstrate that the proposed method is highly accurate and flexible in a large-scale measurement volume and can perfectly meet the requirements of industrial applications. A comparison with current state-of-the-art methods is also conducted to show that the proposed method is competitive in globalization accuracy and superior in flexibility and measurement volume.

In the future, we will further investigate the error theory of the iGPS and optimize the calibration mathematical model to further improve the accuracy and robustness of the local data globalization method.

## Figures and Tables

**Figure 1 sensors-24-06114-f001:**
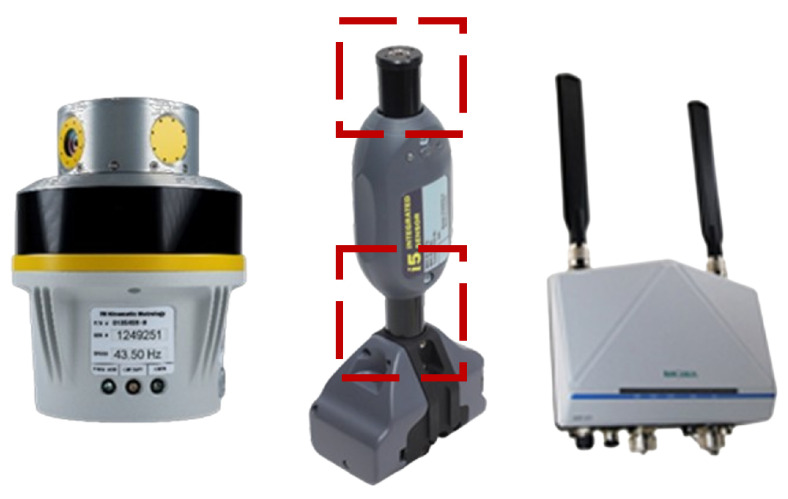
The main components of the iGPS.

**Figure 2 sensors-24-06114-f002:**
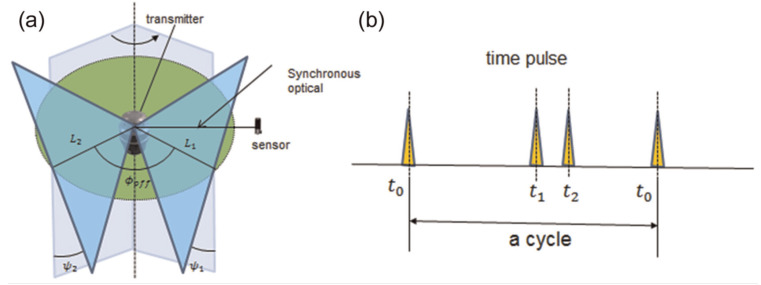
Working principle of a single transmitter (**a**,**b**).

**Figure 3 sensors-24-06114-f003:**
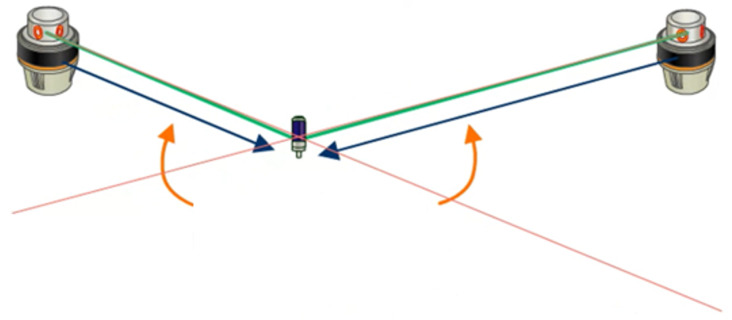
Working principle of multiple transmitters.

**Figure 4 sensors-24-06114-f004:**
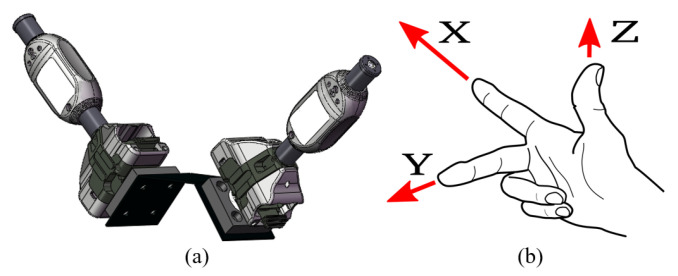
(**a**) Two i5IS sensors used to construct the sensor coordinate system. (**b**) Right-hand coordinate system.

**Figure 5 sensors-24-06114-f005:**
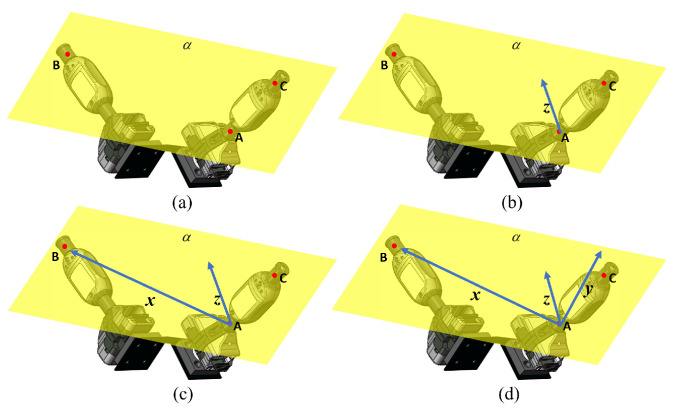
The method for constructing the sensor coordinate system with three points. (**a**) Determine the plane by three using points. (**b**) Determine the original point and the *z*-axis. (**c**) Determine the *x*-axis. (**d**) Determine the *y*-axis.

**Figure 6 sensors-24-06114-f006:**
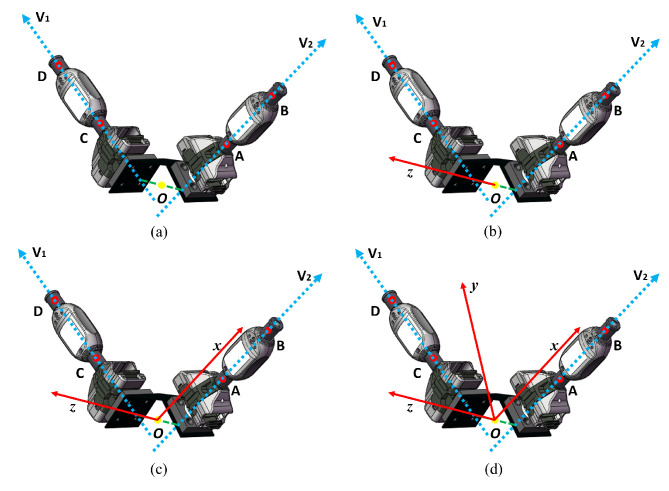
The method for constructing the sensor coordinate system with four points. (**a**) Construct the vector by the using 4 points and determine the original point. (**b**) Determine the *z*-axis. (**c**) Determine the *x*-axis. (**d**) Determine the *y*-axis.

**Figure 7 sensors-24-06114-f007:**
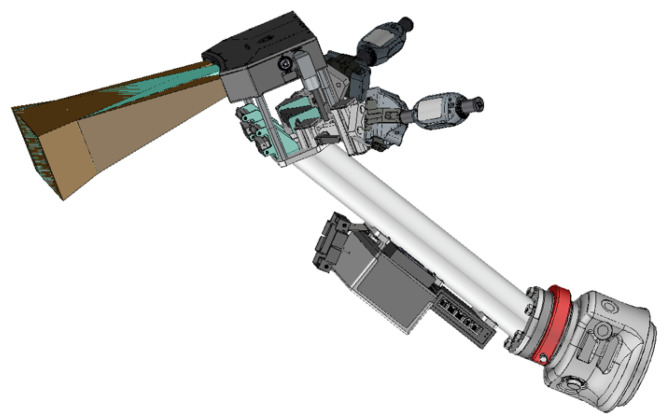
Integrated measurement unit consisting of a 3D scanner and two i5IS sensors.

**Figure 8 sensors-24-06114-f008:**
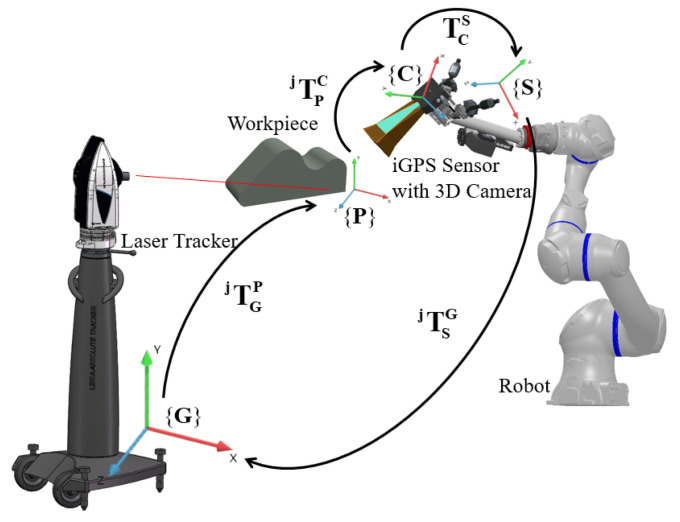
Calibration principle with a laser tracker.

**Figure 9 sensors-24-06114-f009:**
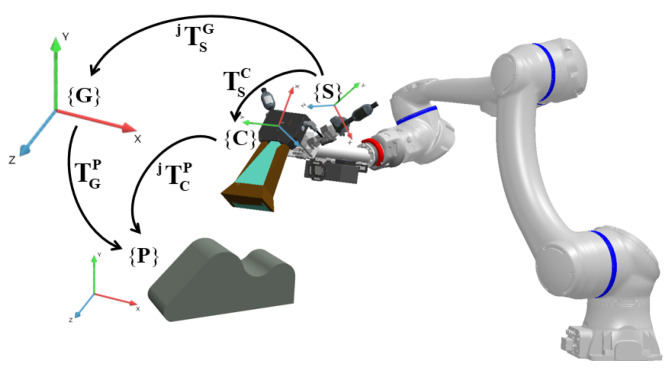
Calibration without a laser tracker.

**Figure 10 sensors-24-06114-f010:**
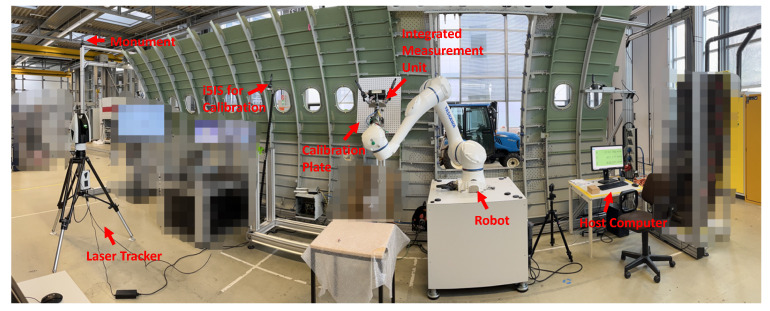
The configuration of the calibration experiment.

**Figure 11 sensors-24-06114-f011:**
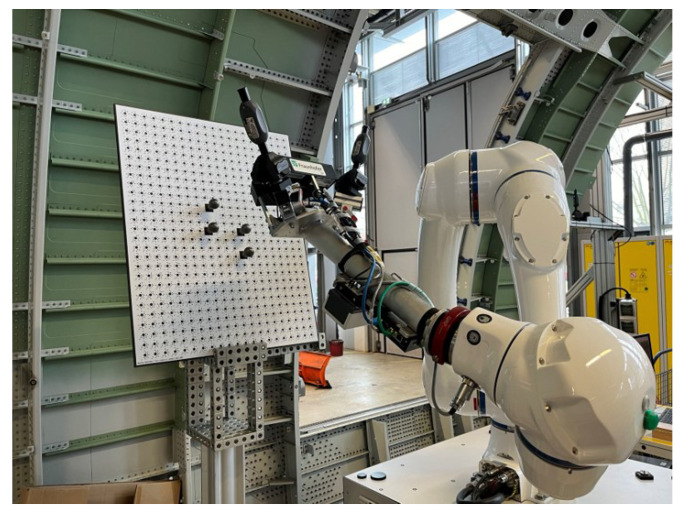
Integrated measurement unit and calibration board.

**Figure 12 sensors-24-06114-f012:**
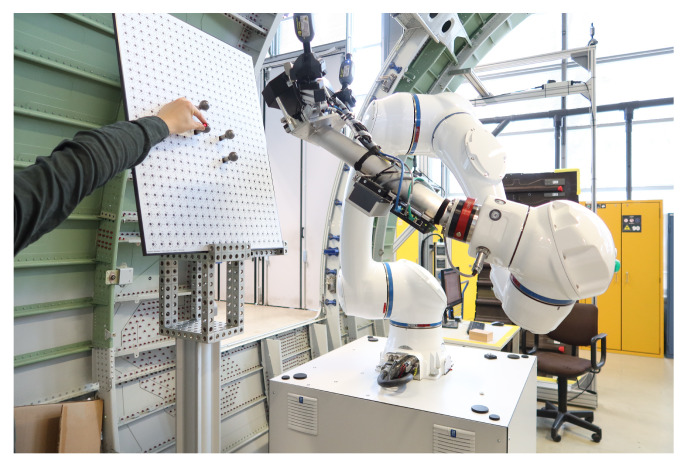
Measure the standard sphere using the SMR as a probe.

**Figure 13 sensors-24-06114-f013:**
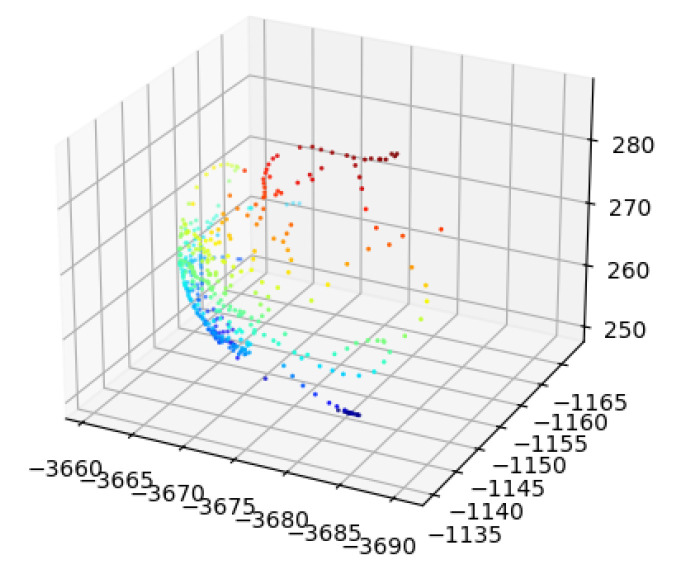
The measurement of the standard sphere using a laser tracker with an SMR.

**Figure 14 sensors-24-06114-f014:**
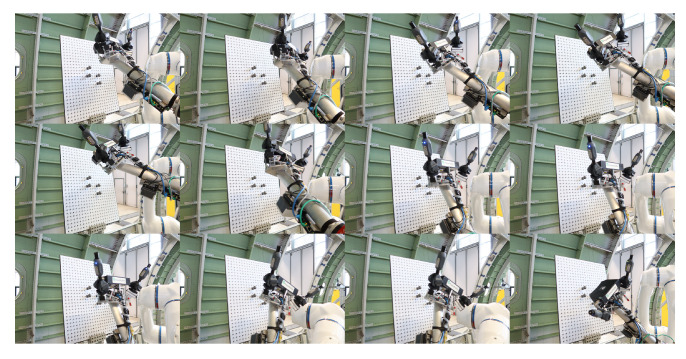
Different relative positions between the integrated measurement unit and the standard sphere.

**Figure 15 sensors-24-06114-f015:**
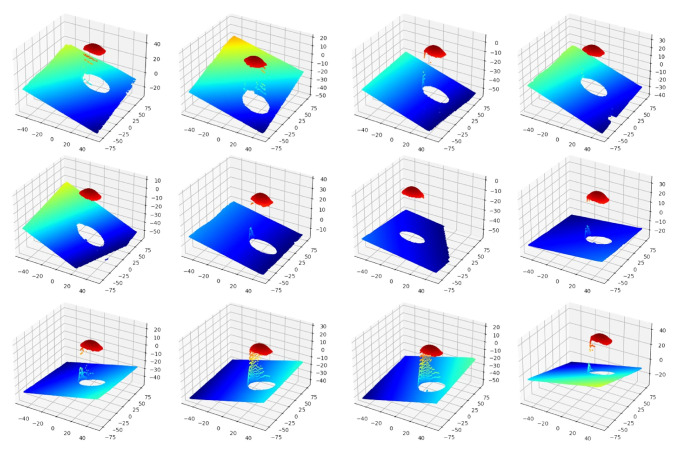
Measurements of the single standard sphere from the 3D scanner.

**Figure 16 sensors-24-06114-f016:**
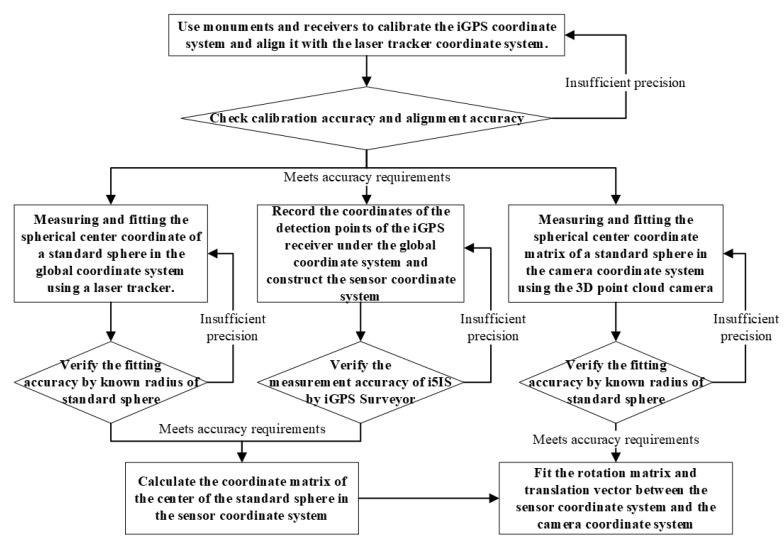
Flowchart of measurement environment calibration using a laser tracker.

**Figure 17 sensors-24-06114-f017:**
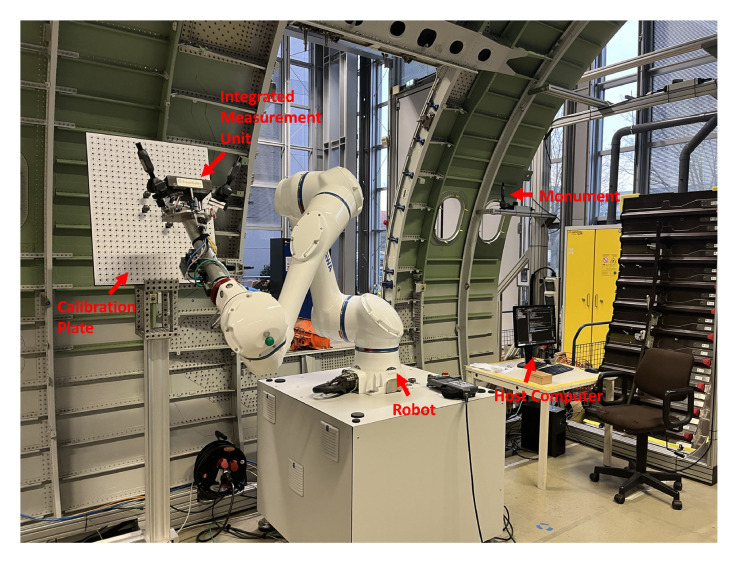
The calibration experiment configuration without using a laser tracker.

**Figure 18 sensors-24-06114-f018:**
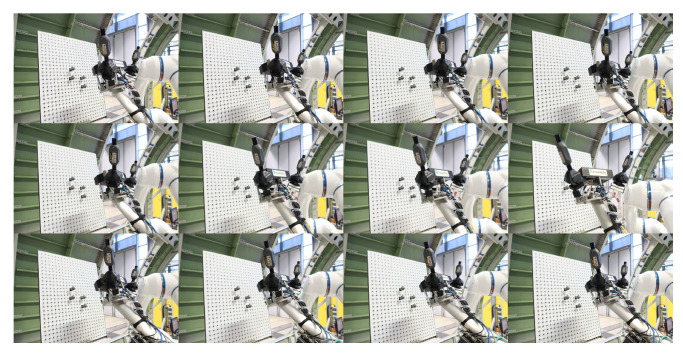
Different relative positions between the integrated measurement unit and the standard sphere during multiple measurements.

**Figure 19 sensors-24-06114-f019:**
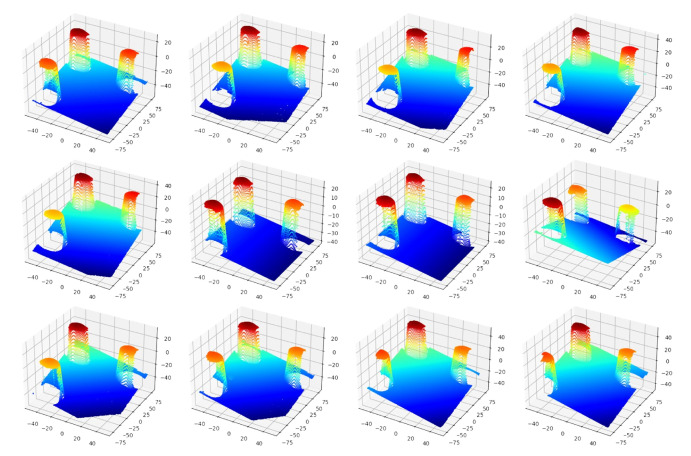
Different measurements of three standard spheres from the 3D scanner.

**Figure 20 sensors-24-06114-f020:**
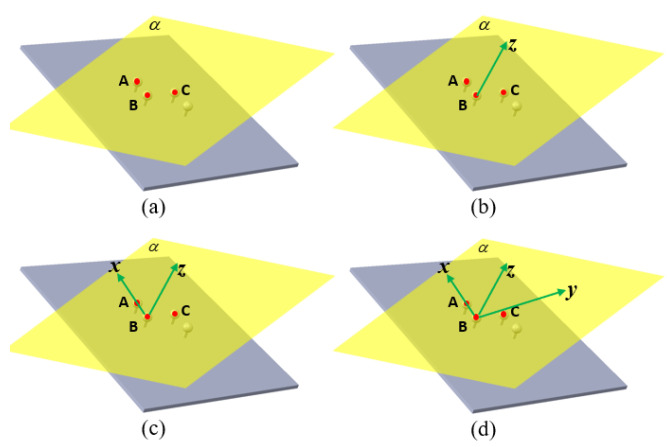
The method for constructing the target coordinate system with three standard spheres. (**a**) Determine the three using standard spheres from four and determine the plane by these three sphere centers. (**b**) Determine the *z*-axis. (**c**) Determine the *x*-axis. (**d**) Determine the *y*-axis.

**Figure 21 sensors-24-06114-f021:**
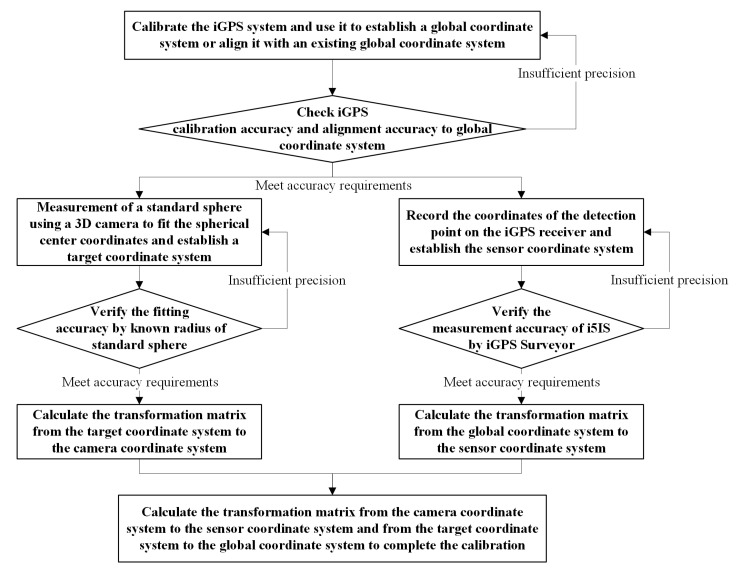
Flowchart of the calibration process without using a laser tracker.

**Figure 22 sensors-24-06114-f022:**
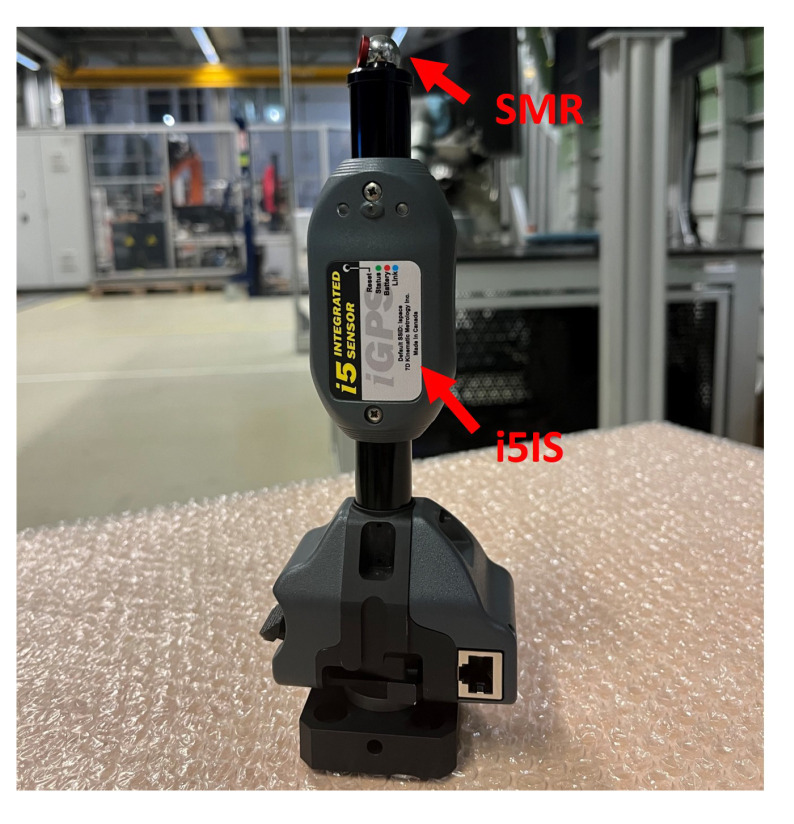
The SMR is placed on the i5IS sensor.

**Figure 23 sensors-24-06114-f023:**
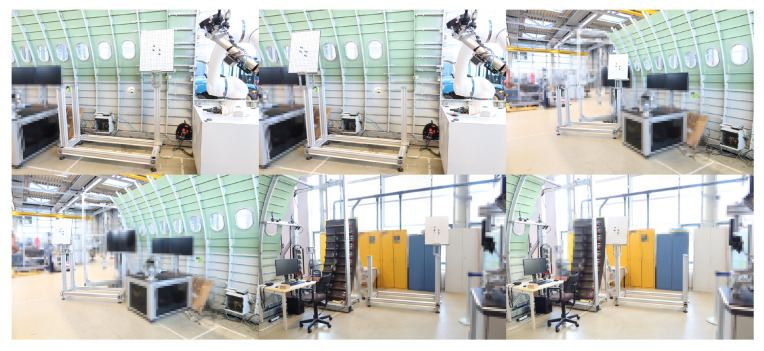
The calibration board is placed in different positions.

**Figure 24 sensors-24-06114-f024:**
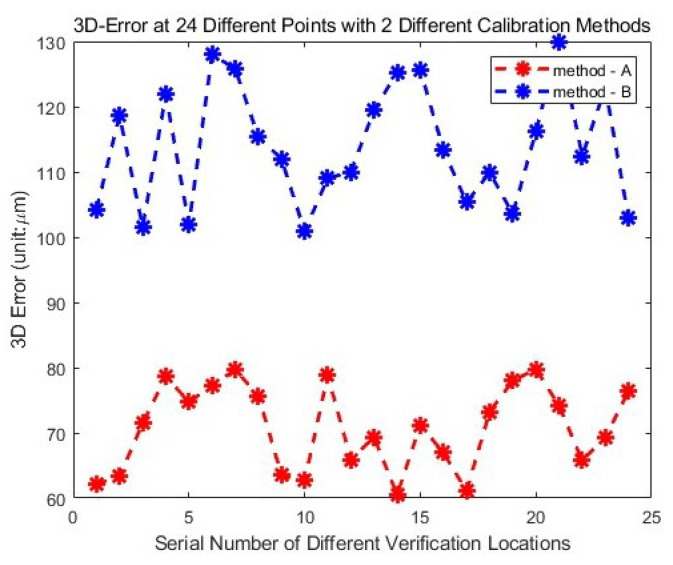
Three-dimensional errors of the two different calibration methods at different positions.

**Figure 25 sensors-24-06114-f025:**
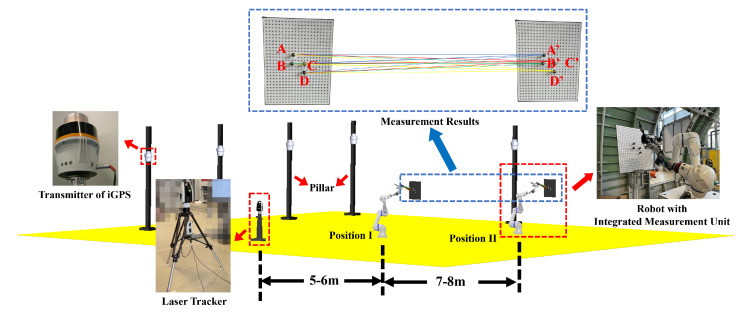
Configuration of Experiment II.

**Figure 26 sensors-24-06114-f026:**
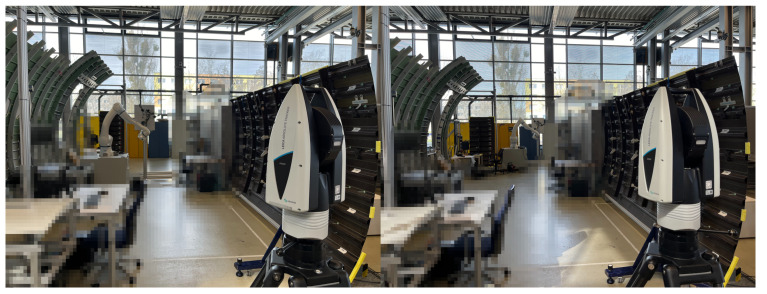
The two positions of the robot in Experiment II.

**Figure 27 sensors-24-06114-f027:**
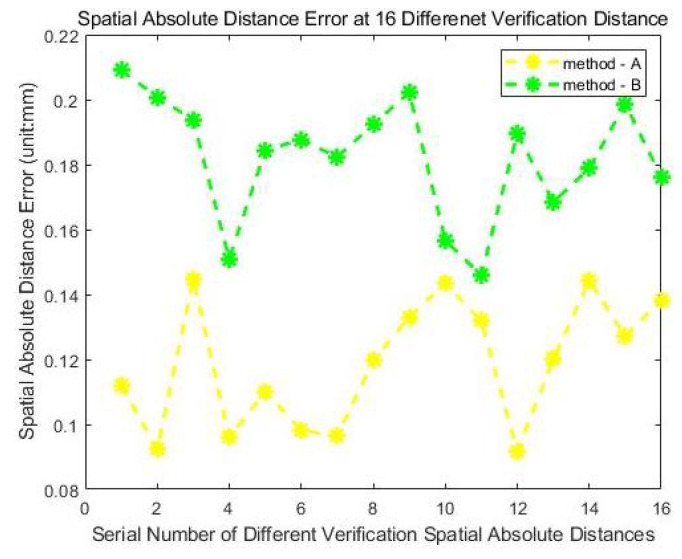
Three-dimensional errors of the two calibration methods at 16 verification distances.

**Figure 28 sensors-24-06114-f028:**
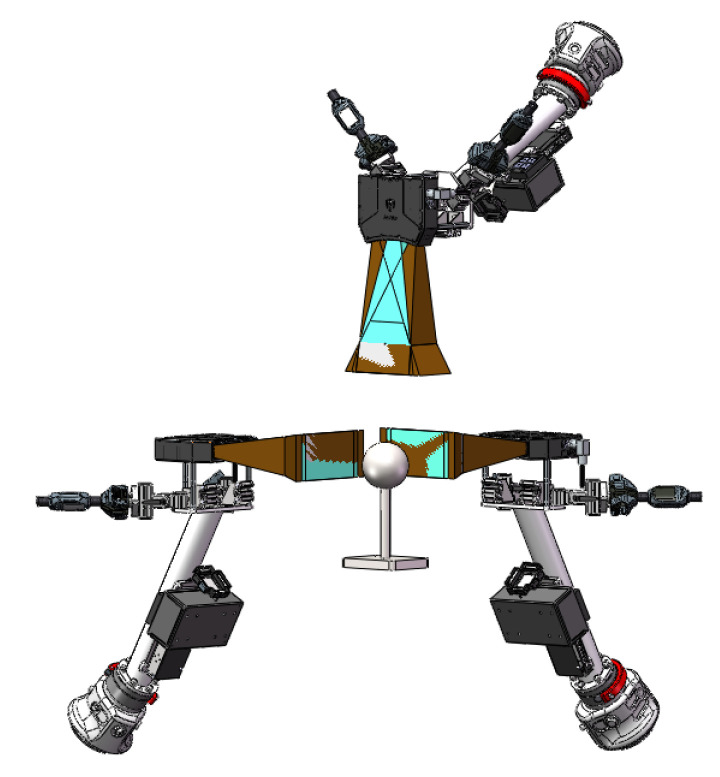
Completely different measurement postures.

**Figure 29 sensors-24-06114-f029:**
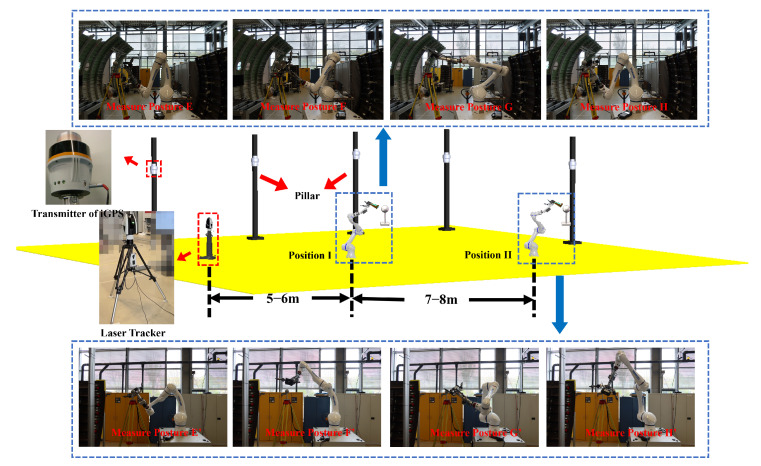
Configuration of Experiment III.

**Figure 30 sensors-24-06114-f030:**
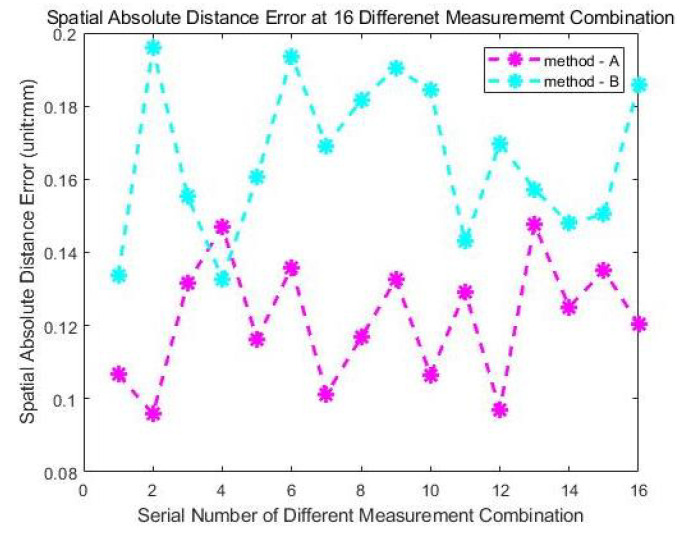
Three-dimensional errors of the two methods with 16 combinations of measurement postures.

**Table 1 sensors-24-06114-t001:** Alignment errors between the two coordinate systems.

$P_{i}$	$x_{e}$ (μm)	$y_{e}$ (μm)	$z_{e}$ (μm)	$3D_{e}$ (μm)
1	20.6046	−48.8167	−22.3077	55.8031
2	−45.3829	−40.2868	32.3458	68.7669
3	19.4829	−18.2901	45.0222	52.3556
4	−46.5554	−6.1256	−11.8442	48.4274
5	26.5517	29.5200	−31.3127	50.5659
6	−1.0236	−5.4414	14.6313	15.6439
7	20.9365	25.4687	−22.3975	39.8577
8	17.9703	15.5098	−33.7388	41.2528
9	−38.1002	−0.1636	45.9744	59.7101
10	−15.9614	8.5268	−27.6188	33.0193

**Table 2 sensors-24-06114-t002:** Statistical error characteristics of the two methods in Experiment I.

Method	${\mathrm{error}}_{\max}$ (μm)	${\mathrm{error}}_{\mathrm{avg}}$ (μm)	STD (μm)
M-A	79.7684	70.8104	6.5407
M-B	129.9847	114.0155	9.2991

**Table 3 sensors-24-06114-t003:** Measurements of 16 spatial distances with three methods.

Notation	LT (mm)	M-A (mm)	M-B (mm)
A-A’	6170.9327	6171.0424	6171.1398
A-B’	6157.8834	6157.7877	6157.6792
A-C’	6166.4338	6166.5748	6166.6237
A-D’	6154.9251	6155.0159	6155.0709
B-A’	6179.0295	6178.9099	6178.8316
B-B’	6165.1125	6165.2082	6165.2977
B-C’	6173.8667	6173.7636	6173.6777
B-D’	6161.6112	6161.7297	6161.8023
C-A’	6162.0308	6162.1629	6162.2323
C-B’	6148.2574	6148.3935	6148.4067
C-C’	6155.4124	6155.2781	6155.2642
C-D’	6143.1247	6143.2118	6143.3095
D-A’	6169.0932	6169.2123	6169.2584
D-B’	6154.5312	6154.3857	6154.3512
D-C’	6161.6974	6161.5629	6161.4912
D-D’	6148.6221	6148.7583	6148.7961

**Table 4 sensors-24-06114-t004:** Statistical error characteristics of two methods in Experiment II.

Method	${\mathrm{error}}_{\max}$ (mm)	${\mathrm{error}}_{\mathrm{avg}}$ (mm)	STD (mm)
M-A	0.1455	0.1189	0.0184
M-B	0.2071	0.1826	0.0196

**Table 5 sensors-24-06114-t005:** Errors of 16 combinations of measurement postures.

Notation	M-A Error (mm)	M-B Error (mm)
E-E’	−0.1066	0.1337
E-F’	0.0958	−0.1959
E-G’	0.1317	0.1554
E-H’	0.1470	0.1328
F-E’	0.1163	0.1605
F-F’	−0.1359	−0.1936
F-G’	−0.1012	0.1692
F-H’	0.1167	−0.1817
G-E’	0.1326	0.1904
G-F’	0.1066	0.1844
G-G’	0.1293	0.1430
G-H’	−0.0971	−0.1699
H-E’	0.1476	−0.1572
H-F’	0.1251	0.1479
H-G’	−0.1351	−0.1504
H-H’	−0.1204	−0.1859

**Table 6 sensors-24-06114-t006:** Statistical error characteristics of the methods in Experiment III.

Method	${\mathrm{error}}_{\max}$ (mm)	${\mathrm{error}}_{\mathrm{avg}}$ (mm)	STD (mm)
M-A	0.1476	0.1216	0.0167
M-B	0.1959	0.1657	0.0212

**Table 7 sensors-24-06114-t007:** Comparison of methods’ effectiveness.

Method	Max Error (mm)	Average Error (mm)	Standard Deviation (mm)	Scale of Space
A. Paoli [5]	3.73	1.6316	1.225	25 m × 5 m
Z. Zhou [6]	0.067	0.064	0.003	ϕ5 m × 7 m
J. Wang [7]	0.187	0.118	0.0529	2800 mm × 877 mm × 360 mm
J. Wang [8]	0.346	0.308	0.346	4.6 m × 2.6 m
S. Yin [10]	0.604	0.309	0.1132	0.9 m × 1.2 m × 0.8 m
M-A Experiment I	0.079	0.070	0.0065	10 m × 10 m × 5 m
M-B Experiment I	0.1299	0.1140	0.0093	10 m × 10 m × 5 m
M-A Experiment II	0.1455	0.1189	0.0184	10 m × 10 m × 5 m
M-B Experiment II	0.2071	0.1826	0.0196	10 m × 10 m × 5 m
M-A Experiment III	0.1476	0.1216	0.0167	10 m × 10 m × 5 m
M-B Experiment III	0.1959	0.1657	0.0212	10 m × 10 m × 5 m

## Data Availability

Data are contained within the article.

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
