# Peer review of "Calibration Methods for Large-Scale and High-Precision Globalization of Local Point Cloud Data Based on iGPS"

_sensors, 2024, doi:10.3390/s24186114_

Round 1

Reviewer 1 Report

Comments and Suggestions for Authors

It is quite an interesting topic. 

1) Accuracy and precision are different in metrology. Please read  

JCGM 200:2012
International Vocabulary of Metrology - Basic and general concepts and associated terms (2012)DOI: https://doi.org/10.59161/JCGM200-2012 (2012)

And develop a better understanding of "accuracy". Many places should use precision (not accuracy)

2) Please give a review of the current calibration method of indoor GPS in section 1, and use a table to compare with your methods.

3) Please remove "WE". The methods should be able to use by anyone. If only the authors can use, it is not suitable to publish.

4) The title of "Research on ...." looks like a review paper. "Calibration methods of ..." may be more suitable.

Comments on the Quality of English Language

English is below average;

Number should use two, three,... (not 2, 3, )

The first letter should capitalised, see line 14, "one" >> "One"

Author Response

Comments 1: [Accuracy and precision are different in metrology. Please read  JCGM 200:2012
International Vocabulary of Metrology - Basic and general concepts and associated terms (2012)DOI: https://doi.org/10.59161/JCGM200-2012 (2012). And develop a better understanding of "accuracy". Many places should use precision (not accuracy)]

Response 1: [Thank you for pointing this out. I agree with this comment. Really thanks for your mention. I have modified these problems properly. I change some explaining ways to remove "we". You can find them exactly at: Page 1 Line 9; Page 2 Line 39; Page 2 Line 47; Page 2 Line 73; Page 3 Line 94; Page 3 Line 106; Page 5 Line 152; Page 6 Line 179; Page 8, Line 223]

Comments 2: [Please give a review of the current calibration method of indoor GPS in section 1, and use a table to compare with your methods.]

Response 2: 

[

 I’d like to give a more detailed explanation about this comment.

  • If you mean the calibration method of the iGPS instrument itself. iGPS is a mature and commercialized industrial measurement device, which has its own calibration method offered by the product instruction. It is meaningless to research about how to calibrate the iGPS itself currently because the calibration method from the manufacturing company can reach high accuracy.
  • If you mean the calibration method of the iGPS in the field of local point cloud data globalization. The use of iGPS as an external measurement system for globalization of local point cloud data is proposed for the first time in this paper, and for the time being there are no other references that propose a similar methodology. In the current introduction, I have listed the other state-of-the-art methods in the field of local point cloud data globalization and the comparison has been executed in the subsection 5.4 and Table 6.
  • If you mean the calibration method of iGPS in the other research fields. There are some works about using iGPS to do some calibration in the other research fields, like calibrate the robot tool center point. However, it is meaningless to be referred according to the topic of this paper.

]

Comments 3: [Please remove "WE". The methods should be able to use by anyone. If only the authors can use, it is not suitable to publish.]

Response 3: [Thank you for pointing this out, I agree with this comment. I have modified this in my manuscript. However, when I am describing the experiment process, “we” only refers to our research team. Because the research and the experiments mentioned in the paper are executed by our research team. You can find my modification exactly at: Page 1 Line 5-6; Page 1 Line 8; Page 1 Line 34-35; Page 2 Line 72; Page 3 Line 98-99; Page 5 Line 144; Page 5 Line 145; Page 6 Line 192-193; Page 7 Line 199; Page 7 Line 216; Page 8 Line 223; Page 8 Line 232; Page 8 Line 235; Page 8 Line 242; Page 8 Line 243; Page 8 Line 234-235; Page 9 Line 246; Page 9 Line 249; Page 11 Line 271; Page 12 Line 287; Page 12 Line 292; Page 12 Line 293; Page 12 Line 297; Page 12 Line 299-300; Page 14 Line 319-320; Page 15 Line 326; Page 16 Line 352; Page 16 Line 354; Page 17 Line 360-361; Page 18 Line 382; Page 20 Line 405; Page 21 Line 415; Page 21 Line 417-418]

Comments 4: [The title of "Research on ...." looks like a review paper. "Calibration methods of ..." may be more suitable.]

Response 4: [Thank you for pointing this out, I agree with this comment. It sounds great. The new title is “Calibration methods of large-scale and high-accuracy globalization of local point cloud data based on iGPS”.]

Reviewer 2 Report

Comments and Suggestions for Authors

This paper proposes a method for transforming point cloud data from multiple targets into a global coordinate system using iGPS. On this basis, methods for establishing models with and without the use of laser trackers for coordinate calibration are proposed, and experiments are designed for demonstration. The manuscript demonstrates a certain level of innovation, with a clear article structure and high credibility. However, there are still the following issues to be addressed:

  1. In section 2.2, the author describes the method of using three and four points to construct the i5IS sensor coordinate system. However, in the experimental section, it is not clearly stated which method was actually used in the experiments. Therefore, the intention of section 2.2 is not clear, and it is uncertain whether there are issues with this design.
  2. There are defects in the design of Experiment 1. The comparison of measurement results from the laser tracker with variables M-A and M-B is not sufficient to prove its impact on the experimental results. It is recommended to optimize the experimental design to ensure the independence of variables, so as to accurately assess the role of the laser tracker.
  3. Some minor errors in the text need to be corrected. For example, "Equation 13" should be written as "Equation 13" in line 309. Line 316 is missing a period.

Comments on the Quality of English Language

Minor editing of English language required.

Author Response

Comments 1: [In section 2.2, the author describes the method of using three and four points to construct the i5IS sensor coordinate system. However, in the experimental section, it is not clearly stated which method was actually used in the experiments. Therefore, the intention of section 2.2 is not clear, and it is uncertain whether there are issues with this design.]

Response 1: [Thank you for pointing this out, I agree with this comment. The main purpose of section 2.2 is to introduce 2 different sensor coordinate system establishment methods using i5IS. In the following experiments, both of two methods have been used to verify that both of two methods can reach high accuracy coordinate system establishment. I have modified this and pointed this out clearly at the end of section 5. You can find out them exactly at Page 21 Line 421-426]

Comments 2: [There are defects in the design of Experiment 1. The comparison of measurement results from the laser tracker with variables M-A and M-B is not sufficient to prove its impact on the experimental results. It is recommended to optimize the experimental design to ensure the independence of variables, so as to accurately assess the role of the laser tracker.]

Response 2:

[

  I am sorry I have not described the process of Experiment 1 clearly, but I feel the design of Experiment I is correct, I’d like to explain it to you more clearly this time.

  1. We use the SMR of the laser tracker to calibrate the iGPS coordinate system. We put the SMR on the magnetic base of the i5IS like the figure below. The main purpose of this step is to align the laser tracker coordinate system and the iGPS coordinate system. Now both of these 2 coordinate systems could be notated as global coordinate system.

  2. We check the alignment accuracy of these two coordinate systems. We place the i5IS with SMR at the different positions in the measurement space, measure the target with iGPS and laser tracker to check the alignment error. We get the table below to prove that the alignment error is much smaller than the measurement error. Therefore, we can prove this experiment is meaningful.

  3. We measure the standard sphere with laser tracker. We use the SMR of laser tracker as a probe to measure the standard sphere on the calibration board (shown as the figure below). We use these measurements from laser tracker to fit the coordinate of the sphere center in the global coordinate system.
  4. We measure the same standard sphere with the integrated measurement unit (shown as the figure below). We first get the measurement (point cloud) from the 3D scanner, fit the coordinate of the sphere center in the 3D scanner coordinate system through this point cloud. Then we transfer this coordinate from the 3D scanner coordinate system to the global coordinate system by the transformation matrix which we calibrate before.
  5. The calibration error is quantified by comparing the Euclidean distance of the spherical center coordinates in the global coordinate system measured by the two measurement methods "M-A” and “M-B” are the notation of calibration methods instead of variables. “M-A” means the calibration method which needs to use laser tracker. “M-B” means the calibration method which doesn’t need to use laser tracker.

I hope I have explained this experiment more clearly this time. I have also added some explanation in my manuscript.

]

Comments 3: [Some minor errors in the text need to be corrected. For example, "Equation 13" should be written as "Equation 13" in line 309. Line 316 is missing a period.]

Response 3: [Thank you for pointing this out, I agree with this comment. You can find them out exactly at Page 13 Line 309; Page 14 Line 316.]

Round 2

Reviewer 1 Report

Comments and Suggestions for Authors

This manuscript is a technical note which pays attention to data collection and analysis via currently available instruments and methods. 

Novelty is enough to be published as a research paper. 

Comments on the Quality of English Language

Please use the number correctly. 

2 methods >> Two methods.

3 >> Three

Author Response

Comments: [This manuscript is a technical note which pays attention to data collection and analysis via currently available instruments and methods. Novelty is enough to be published as a research paper.Please use the number correctly. 2 methods >> Two methods. 3 >> Three]

Response: [Thank you for pointing this out. I agree with you. You can find them exactly at the highlight parts of the new version of manuscript]